# Zero-Shot Tokenizer Transfer

**Benjamin Minixhofer** [SEP]    **Edoardo M. Ponti** [CLS]    **Ivan Vulić** [SEP]

[SEP] University of Cambridge        [CLS] University of Edinburgh

## Abstract

Language models (LMs) are bound to their tokenizer, which maps raw text to a sequence of vocabulary items (*tokens*). This restricts their flexibility: for example, LMs trained primarily on English may still perform well in other natural and programming languages, but have vastly decreased efficiency due to their English-centric tokenizer. To mitigate this, we should be able to swap the original LM tokenizer with an arbitrary one, on the fly, without degrading performance. Hence, in this work we define a new problem: Zero-Shot Tokenizer Transfer (ZeTT). The challenge at the core of ZeTT is finding *embeddings* for the tokens in the vocabulary of the new tokenizer. Since prior heuristics for initializing embeddings often perform at chance level in a ZeTT setting, we propose a new solution: we train a hypernetwork taking a tokenizer as input and predicting the corresponding embeddings. We empirically demonstrate that the hypernetwork generalizes to new tokenizers both with encoder (e.g., XLM-R) and decoder LLMs (e.g., Mistral-7B). Our method comes close to the original models' performance in cross-lingual and coding tasks while markedly reducing the length of the tokenized sequence. We also find that the remaining gap can be quickly closed by continued training on less than 1B tokens. Finally, we show that a ZeTT hypernetwork trained for a base (L)LM can also be applied to fine-tuned variants without extra training. Overall, our results make substantial strides toward detaching LMs from their tokenizer.

## 1    Introduction

Language Models[1] typically operate on discrete tokens, so they need a means to map text into a sequence of tokens, namely a *tokenizer*. The vast majority of contemporary LMs use subword tokenizers (Devlin et al., 2019; Jiang et al., 2023; Touvron et al., 2023; Parmar et al., 2024, among others), whereas others use byte-level (Xue et al., 2022; Yu et al., 2023; Wang et al., 2024) or character-level tokenizers (Clark et al., 2022; Tay et al., 2022). Regardless of the chosen tokenization 'granularity', these models share a fundamental limitation: once they are trained with a particular tokenizer, inference with a different tokenizer is impossible. In other terms, a pre-trained LM is *"bound"* to the tokenizer it was trained with. This has wide-ranging implications: since the focus during pretraining is typically primarily on the English language, the tokenizer often encodes languages besides English (Rust et al., 2021) or other domains, such as code, less efficiently. This leads to large disparities in the inference cost between English and non-English text (Ahia et al., 2023; Petrov et al., 2023). Tokenizers may also be sub-optimal for domains which they were not designed to be used with, e.g. fine-tunings of the Llama models performing subpar on coding tasks (Dagan et al., 2024). Efficiency and performance are only some of the reasons to transfer models across tokenizers: methods of interaction between models, such as ensembling (Sagi & Rokach, 2018) and model merging (Wortsman et al., 2022; Ainsworth et al., 2023; Yadav et al., 2023), typically assume the same unit of representation (i.e., equivalent tokenization) across models; if two models adopt different

---

[1] We adopt a broad definition of LMs that also includes models that do not define a probability distribution over finite-length sequences, such as text encoders.

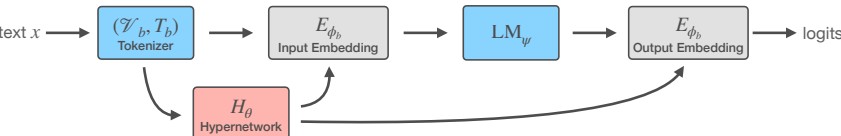

Figure 1: The hypernetwork predicts input and output embeddings based on the tokenizer.

tokenizers, they become unsuitable for ensembling or merging. Problematic artifacts of tokenization such as 'Glitch tokens' (Land & Bartolo, 2024) may also be fixed via transfer to a new tokenizer.

To address these issues, past work developed methods to equip an LM with a new tokenizer by retraining the embedding parameters, and optionally continuing to train the entire model (Artetxe et al., 2020; de Vries & Nissim, 2021). This adaptation can be made faster by initializing the embedding parameters through heuristics (Tran, 2020; Minixhofer et al., 2022; Gee et al., 2022; Dobler & de Melo, 2023; Liu et al., 2023). In this work, we formulate a new problem: given an LM, can we create an embedding matrix on-the-fly for any arbitrary tokenizer, without ever observing data for it? While past work investigated $n$-shot tokenizer transfer, we refer to this new problem as *zero-shot tokenizer transfer* (ZeTT). If the performance of the model can be approximately preserved, ZeTT effectively "detaches" LMs from the tokenizer they were trained with. We first evaluate the efficacy of prior (heuristic-based) approaches for ZeTT, finding that, while heuristics can preserve performance to some extent, there is generally a large gap to the original LM performance.

To close this gap, we introduce a new paradigm: We train a *hypernetwork* on a diverse distribution of tokenizers to predict the embedding parameters for any given tokenizer. By investing in the one-time cost of training the hypernetwork, we aim to subsequently enable effective ZeTT. This proves to be possible: ZeTT via the hypernetwork preserves performance to a few percent accuracy in many cases. Furthermore, the hypernetwork can learn to rapidly adapt to a given target tokenizer by continued training on a small amount (<1B) of extra tokens, whereas previous work typically needed hundreds of billions of tokens (Dagan et al., 2024). As such, our hypernetwork provides a state-of-the-art solution to $n$-shot tokenizer transfer, while also establishing a competitive baseline for our newly introduced zero-shot tokenizer transfer problem. This unlocks a range of new ways to combine language models with tokenizers. For example, in this work, we zero-shot substitute the Mistral-7B tokenizer (Jiang et al., 2023) with a tokenizer that encodes code using 10% fewer tokens on average, while preserving functional code generation correctness to approx. 3% (Section 4.2). We also evaluate zero-shot cross-lingual transfer of the multilingual XLM-R encoder model to a range of different languages by substituting the XLM-R tokenizer with a target-language specific tokenizer and reusing adapters trained for the original XLM-R. This leads to a >16% speedup and preserves performance on XNLI (Conneau et al., 2018) to 1% on average. Finally, we show that a hypernetwork trained for a base large LM (e.g. Mistral-7B) can also be applied to fine-tunings of the same model (e.g. Mistral-7B-Instruct-v0.1), preserving capabilities to a large extent (Section 4.3).

## 2 Background

**Tokenizers and Embeddings.** Tokenizers operate as a *tokenization function $T$* mapping a text to a sequence of elements in the *vocabulary $\mathcal{V}$*. By the term *tokenizer*, we henceforth refer to the tuple comprising the two crucial components, $(\mathcal{V}, T)$. Importantly, the vocabulary and the tokenization function are distinct components; given some vocabulary, there are many ways to encode text as a sequence of tokens in this vocabulary (e.g. Hofmann et al., 2022; Uzan et al., 2024). After tokenization, the model represents the sequence of tokens via a function $E_\phi : \mathcal{V} \to \mathbb{R}^{d_\mathrm{model}}$ (the *embeddings*). The embeddings are typically parametrized by a matrix $\phi$ as a lookup table which assigns a distinct $d_\mathrm{model}$-dimensional vector (a row of the matrix) to every element in $\mathcal{V}$. Embeddings are used twice in the language model: once at the input to map tokens to a fixed-size vector, and again at the output to compute a logit for every token, typically via a dot-product of $E_\phi(t)$ with the final hidden state of the LM. Embedding parameters may or may not be shared between the input and the output;[2] our method works with both. We denote the entire set of embedding parameters via $\phi$, denoting input embeddings as $\phi^\mathrm{in}$ and output embeddings as $\phi^\mathrm{out}$, if necessary.

---

[2]Some models share the input and the output embedding parameters (e.g. Conneau et al., 2020), this has been shown to be problematic (Chung et al., 2021) and many recent LLMs (e.g. Jiang et al., 2023) separate them.

Contemporary language models typically use subword tokenizers via BPE (Sennrich et al., 2016) or UnigramLM (Kudo, 2018). Subword tokenization is a common choice since it can represent arbitrary sequences of text ("open-vocabulary" language modeling) while largely retaining the efficiency of word-level models (Mielke et al., 2021). However, there are a number of problems with the (lack of) robustness of subword tokenization (Xue et al., 2022; Golkar et al., 2023). A recent strand of work aims to get rid of subword tokenization via byte-level (so-called "token-free") models (Xue et al., 2022; Yu et al., 2023). However, these models still operate on tokens, using the set of 256 bytes as the vocabulary, and UTF-8 as the tokenization function (Mielke et al., 2021). In a similar vein, some models use character-level tokenization (Tay et al., 2022; Clark et al., 2022), optionally learning to pool characters into longer tokens (Nawrot et al., 2023). So far, byte- or character-level approaches have been unable to supplant subword tokenization due to longer sequences resulting in higher compute requirements, and not necessarily being more robust (Libovický et al., 2022). Thus, although our approach is applicable to any tokenizer, we focus our experiments on subword tokenizers. Specifically, we use the UnigramLM parametrization of the tokenization function, and show that other tokenizers can be converted to this parametrization later in Section 5. UnigramLM sets $T(x) := \operatorname{argmax}_{C \in \mathcal{C}_x} \sum_{t \in C} \log p(t)$ where $\mathcal{C}_x$ is the set of all possible decompositions of $x$ in $\mathcal{V}$. This provides a convenient way to represent tokens as a 2-tuple $(t, p(t)) \in (\mathcal{V}, \mathbb{R})$.

**Embedding Initialization Heuristics.** Prior work transfers LMs to a new tokenizer by initializing embedding parameters via a heuristic, then continuing to train the embeddings. We denote the original tokenizer as $(\mathcal{V}_a, T_a)$ and the original embedding parameters as $\phi_a$. Analogously, the target tokenizer is $(\mathcal{V}_b, T_b)$ with embedding parameters $\phi_b$. FVT (Gee et al., 2022) initializes embeddings for any new token $t \in \mathcal{V}_b$ as the mean of the embeddings of $T_a(t)$ i.e. the mean of the sequence of embeddings the new token is decomposed into by the previous tokenizer $T_a$. RAMEN (Tran, 2020), WECHSEL (Minixhofer et al., 2022) and OFA (Liu et al., 2023) require auxiliary embeddings $E_{\text{aux}} : \mathcal{V}_{\text{aux}} \to \mathbb{R}^{d_{\text{aux}}}$ with $|\mathcal{V}_{\text{aux}} \cap \mathcal{V}_a| \ll |\mathcal{V}_a|$ and $|\mathcal{V}_{\text{aux}} \cap \mathcal{V}_b| \ll |\mathcal{V}_b|$. They use $E_{\text{aux}}$ to embed tokens in $\mathcal{V}_a$ and $V_b$ in the same semantic space, then initialize embeddings in $E_{\phi_b}$ as a weighted average of embeddings in $E_{\phi_a}$ with weights given by their similarity in $E_{\text{aux}}$. FOCUS (Dobler & de Melo, 2023) initializes embeddings of tokens in $V_b \setminus V_a$ as a weighted combination of the overlapping tokens $V_a \cap V_b$, and copies the embeddings of the overlapping tokens. Weights are again computed using an auxiliary embedding matrix $E_{\text{aux}}$, but the only requirement is $|\mathcal{V}_{\text{aux}} \cap \mathcal{V}_b| \ll |\mathcal{V}_b|$. We use FOCUS as the main baseline since Dobler & de Melo (2023) show it obtains better performance without any training (i.e., zero-shot) than other heuristics, which we also confirm later in Section 4.2.

**Heuristic-Free Tokenizer Transfer.** In addition to heuristics, there is also research into changing the training procedure to facilitate $n$-shot tokenizer transfer. Marchisio et al. (2023) show that forward- and backward-propagating through a subset of the model layers is sufficient for learning embeddings for a new tokenizer. Chen et al. (2023) find that regularly resetting the embedding parameters during pretraining boosts the speed at which they are relearnt upon transfer. These approaches can be seen as orthogonal to ours. They could be freely combined with our method; we leave this to future work.

**Embedding Prediction Hypernetworks.** Hypernetworks are networks that predict the parameters of another network (Ha et al., 2017). Prior work uses hypernetworks to predict embeddings for out-of-vocabulary (Pinter et al., 2017) or rare words (Schick & Schütze, 2019, 2020) of word embedding models (Mikolov et al., 2013) and BERT (Devlin et al., 2019). In contrast, our hypernetwork (i) approaches the more general problem of *transferring* to an arbitrary tokenizer, instead of *extending* the original tokenizer and (ii) can be applied to encoder and decoder LMs, that is, it is *objective-agnostic*.

## 3 Methodology

### 3.1 Hypernetwork Training

We aim to find parameters $\theta$ of a hypernetwork $H_\theta : (\mathcal{V}_b, T_b) \to \phi_b$ for some pretrained LM. Let $\phi_a$ and $\psi$ be the embedding and inner (non-embedding) parameters of the language model, respectively. $\mathcal{L}$ is the loss of the language model as a function of the tokens, the embedding parameters, and the inner parameters, typically:

$$\mathcal{L}(t, \phi_a, \psi) = \operatorname{CrossEntropy}(\operatorname{LM}_\psi(E_{\phi_a}(t)), \operatorname{label}(t)),$$

where $\operatorname{LM}_\psi$ is the language model and $\operatorname{label}$ maps the sequence of tokens to corresponding labels, e.g., shifting the sequence in case of standard (autoregressive, causal) language modeling, or masking

---
**Algorithm 1** Hypernetwork training loop for Zero-Shot Tokenizer Transfer
---
  **Input**: corpus $\mathcal{D}$, tokenizer sample size $n$, batch size $m$, max. token length $l$, vocabulary size $k$, noise parameters $(\mu, \sigma)$, pretrained LM parameters $\psi$, initial hypernetwork parameters $\theta_{\text{init}}$.
  **Output**: Hypernetwork parameters $\theta$.

1: **procedure** TRAINHYPERNETWORK
2:   $\theta \leftarrow \theta_{\text{init}}$
3:   $\boldsymbol{q} \leftarrow \text{queue}(x_1, .., x_n \sim \mathcal{D})$ ▷Create a pool of $n$ texts (where $n \geq m$).
4:
5:   **for** step in train_steps **do**
6:     $x_1, .., x_m \sim \mathcal{D}$
7:     $\boldsymbol{q} \leftarrow \text{pop}(\boldsymbol{q}, m)$ ▷Remove the least-recently-added batch.
8:     $\boldsymbol{q} \leftarrow \text{push}(\boldsymbol{q}, x_1, .., x_m)$ ▷Add the current batch.
9:
10:     $\boldsymbol{t}, \boldsymbol{f} \leftarrow \text{substrings}(\boldsymbol{q}, l)$ ▷Compute all substrings and their frequency in $\boldsymbol{q}$.
11:     $\boldsymbol{f} \leftarrow \boldsymbol{f} / \sum_i f_i$ ▷Normalize frequencies to sum to one.
12:     $z \sim \text{Lognormal}(\mu, \sigma^2)$
13:     **for** $t, f \in (\boldsymbol{t}, \boldsymbol{f})$ **do**
14:       $p(t) \leftarrow f + \mathcal{N}(0, z^2)$ ▷Assign a score based on frequency + noise to the substrings.
15:     Sort $\boldsymbol{t}$ by $p(t)$ descending.
16:     $\mathcal{V}_b \leftarrow \boldsymbol{t}[: k]$ ▷Assemble the top $k$ substrings into the tokenizer.
17:     $T_b \leftarrow \text{UnigramLM}(\{(t, p(t)) \mid t \in \boldsymbol{t}[: k]\})$
18:
19:     $\text{loss} \leftarrow \mathcal{L}_\theta(T_b(\boldsymbol{x}), H_\theta(\mathcal{V}_b, T_b), \psi)$ ▷Compute the loss on the $m$ texts in the current batch.
20:     update $\theta$ using $\nabla \theta$ w.r.t. loss.
---

the sequence in case of Masked Language Modeling (Devlin et al., 2019). Importantly, however, we do not make any specific assumptions on $\mathcal{L}$.

Note that the loss of the language model under the original tokenizer $T_a$ on a text $x$ is $\mathcal{L}(T_a(x), \phi_a, \psi)$. We train our hypernetwork to minimize the loss $\mathcal{L}_\theta(T_b(x), H_\theta(\mathcal{V}_b, T_b), \psi)$. That is, we substitute the original embedding parameters for the hypernet predictions, and substitute the original tokenizer for a tokenizer $(\mathcal{V}_b, T_b)$. Figure 1 illustrates the flow of information.

**Defining Distributions over Texts and Tokenizers.** We follow standard practice and sample texts uniformly from the training corpus. Tokenizer sampling is not as trivial: we would like a distribution over tokenizers $(\mathcal{V}_b, T_b)$ with high variance to encourage generalization to unseen tokenizers. To this end, we introduce a procedure to sample a diverse set of UnigramLM tokenizers. We show later in Section 5 that arbitrary tokenizers can be well-approximated via UnigramLM, motivating this choice.

We initially fill a queue $\boldsymbol{q}$ with $n$ texts sampled randomly from the training corpus and, at every step in the training loop, push the $m$ texts in the current batch and remove the $m$ least recently added texts. We then compute all substrings $t$ up to length $l$ and their frequency in $\boldsymbol{q}$.[3][4] We add Gaussian noise to the frequencies to arrive at a final score $p(t)$ for every token $t$. Finally, we assemble the tokenizer by taking the top $k$ tokens with the highest $p(t)$ as the vocabulary and UnigramLM parametrized by $p(t)$ as the tokenization function. The training loop is summarized in Algorithm 1. The 'rolling' queue of texts $\boldsymbol{q}$ ensures high variance in the vocabulary, while the Gaussian noise added to the frequencies ensures high variance in the tokenization function.

Importantly, the texts and the tokenizer are sampled *dependently*: the batch of $m$ texts used for training is a subset of the $n$ texts used for sampling the tokenizer. If they were sampled independently, the probability for a token to occur would be $p(\text{token}) \propto p(\text{token} \in \mathcal{V}_b) \times p(\text{token} \in \boldsymbol{x})$. Since both these factors are small for rare tokens, $p(\text{token})$ would get vanishingly small in this case.

**MIMICK-Style Warmup & Auxiliary Loss.** In practice, directly minimizing $\mathcal{L}_\theta$ starting from randomly initialized $\theta$ is difficult. Thus, we include a warmup stage where we train the hypernetwork to mimic the embedding parameters of the original tokenizer, akin to MIMICK (Pinter et al., 2017).

$$\mathcal{L}_\theta^{\text{warmup}} = \|H_\theta(\mathcal{V}_a, T_a) - \phi_a\|_2$$

---
[3]In practice, implementing $\boldsymbol{q}$ as a queue allows efficiently caching the substrings and their probability $p(t)$ at this step. They only need to be recomputed for the new $m$ texts encountered in every batch.

[4]To ensure substrings do not cross word boundaries we pretokenize the text before computing substrings.

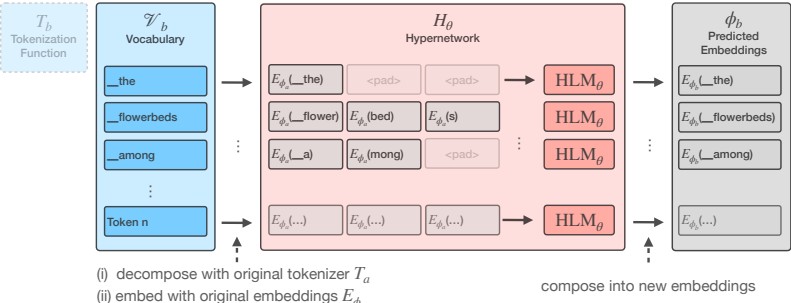

(i) decompose with original tokenizer $T_a$
(ii) embed with original embeddings $E_{\phi_a}$

compose into new embeddings

Figure 2: The hypernetwork consists of a language model $\text{HLM}_\theta$ learning to compose embeddings under the original tokenization into a new embedding and amortizes over the tokenization function.

The warmup stage is substantially quicker than the main stage because there is no need to propagate through the main model. We found it prevents divergence in some cases. Afterwards, we add an auxiliary loss, which, for every token in the sampled vocabulary $\mathcal{V}_b$ that also exists in the original vocabulary $\mathcal{V}_a$, penalizes the distance to the corresponding embedding in $\phi_a$.

$$\mathcal{L}_\theta^{\text{aux}} = \frac{1}{|\mathcal{V}_a \cap \mathcal{V}_b|} \sum_{t \in |\mathcal{V}_a \cap \mathcal{V}_b|} \|H_\theta(\mathcal{V}_b, T_b)[\mathcal{V}_b[t]] - \phi_a[\mathcal{V}_a[t]]\|_2$$

This penalizes drift from the warmup stage. Combining it with the main loss yields the final loss.

$$\mathcal{L}_\theta^{\text{final}} = \mathcal{L}_\theta(T_b(x), H_\theta(\mathcal{V}_b, T_b), \psi) + \alpha \cdot \mathcal{L}_\theta^{\text{aux}}$$

The hyperparameter $\alpha$ weighs the contribution of the auxiliary loss. Since $H_\theta(\mathcal{V}_b, T_b)$ is also required for the main loss, it requires negligible extra computation. The auxiliary loss is necessary especially for models with separate input and output embedding matrices as shown in Appendix B.

### 3.2 Hypernetwork Architecture

It remains to define the hypernetwork architecture, that is, how to map the tokenizer $(\mathcal{V}_b, T_b)$ to the embedding parameters $\phi_b$. To this end, we represent the new tokens $t_b \in \mathcal{V}_b$ by decomposing them using the original tokenization function $T_a$, and embedding them with the original embeddings $E_{\phi_a}$.[5] This sequence of embeddings is passed through multiple Transformer layers, plus a separate prediction head for the input embeddings and output embeddings $\phi_b^{\text{in}}$ and $\phi_b^{\text{out}}$. The hypernetwork thus consists of *another language model* which is applied separately for every token. We refer to the hypernetwork's language model as $\text{HLM}_\theta$. $\text{HLM}_\theta$ can be thought of as learning how to compose the sequence of tokens $T_a(t)$—which any given token is decomposed into—into one embedding, as illustrated in Figure 2. Importantly, we do not take the tokenization function into account. By sampling diverse tokenizers during the training process, we aim for the hypernetwork to learn to produce a single embedding suitable to a wide variety of different tokenization functions. We analyze the impact of this choice later in Section 5. We also experiment with hypernetworks which do take the tokenization function into account in Appendix C.

**On Token Decomposition.** The input to the hypernetwork consists of the sequence of tokens $T_a(t)$ that any given token is *decomposed* into. However, this decomposition is not always trivial: for example, $T_a$ could be character-level, while the token $t$ could be in the vocabulary of a byte-level tokenizer $T_b$. In this case, $t$ could be any arbitrary sequence of bytes (not necessarily valid UTF-8). To solve this issue, we introduce a procedure to convert tokenizers to the byte level by adding a small amount of extra tokens to the vocabulary (c.f. Section 5). This guarantees that $T_a$ can decompose arbitrary tokens. The embeddings of the extra vocabulary are initialized randomly and trainable alongside the hypernetwork parameters.

---

[5]In the multilingual case, we also append an element containing a learnable language-specific embedding.

Table 1: Accuracy on XNLI when *reusing* adapters trained for the original XLM-R model with new zero-shot transferred language-specific tokenizers. Also shown are the absolute change in accuracy from applying our hypernetwork (Δaccuracy) and the average decrease in token length of the language-specific tokenizers over the original tokenizer (Δlength).

|  | ar | bg | de | el | en | es | fr | hi | ru | sw | tr | ur | vi | Avg. |
|---|---|---|---|---|---|---|---|---|---|---|---|---|---|---|
| original | 68.9 | 75.6 | 74.7 | 73.7 | 82.3 | 76.9 | 76.8 | 68.4 | 72.9 | 63.5 | 72.2 | 64.7 | 73.1 | 72.6 |
| Lexical | 58.7 | 63.1 | 65.3 | 61.7 | 72.8 | 68.4 | 66.7 | 61.8 | 62.3 | 51.8 | 58.5 | 60.0 | 72.0 | 63.3 |
| FVT | 63.9 | 70.3 | 70.9 | 67.4 | 79.0 | 73.9 | 71.9 | 65.7 | 67.8 | 57.1 | 66.3 | 61.7 | 72.9 | 68.4 |
| OFA | 57.3 | 64.2 | 67.3 | 62.8 | 73.6 | 68.6 | 68.4 | 61.8 | 63.1 | 54.8 | 59.7 | 59.3 | 72.3 | 64.1 |
| FOCUS | 64.8 | 71.0 | 71.6 | 67.7 | 79.6 | 74.4 | 72.6 | 64.5 | 68.1 | 55.7 | 67.3 | 61.9 | 72.6 | 68.6 |
| ours | **67.9** | **73.9** | **74.1** | **71.4** | **81.1** | **76.2** | **74.7** | **67.7** | **70.7** | **62.3** | **68.7** | **63.2** | **73.9** | **71.2** |
| Δaccuracy | -1% | -2% | -1% | -2% | -1% | -1% | -2% | -1% | -2% | -1% | -3% | -2% | +1% | -1% |
| Δlength | -22% | -14% | -13% | -23% | -9% | -11% | -12% | -13% | -13% | -19% | -15% | -9% | -3% | -14% |

Table 2: Performance of Mistral-7B-v0.1 after zero-shot and $n$-shot tokenizer transfer (training on 800M tokens). We evaluate transfer to the GPT2 tokenizer on natural language benchmarks and transfer to the StarCoder tokenizer on HumanEvalPack. Note that continued training with the original tokenizer (*original@800M*) does not consistently improve performance.

| #shots | Method | Natural Language ($\rightarrow$ GPT2 Tok.) | | | | | | Code (pass@1) ($\rightarrow$ StarCoder Tok.) | | | | | |
|---|---|---|---|---|---|---|---|---|---|---|---|---|---|
| | | | | | | | | | HumanEvalPack | | | | |
| | | PiQA | HS | ARC | BoolQ | MMLU | Avg. | js | go | py | cpp | java | Avg. |
| original | | 80.7 | 81.0 | 79.5 | 83.6 | 59.6 | 76.9 | 28.7 | 20.1 | 29.3 | 29.9 | 32.3 | 28.1 |
| original@800M | | 82.1 | 82.7 | 80.6 | 80.6 | 57.8 | 76.8 | 31.7 | 19.5 | 28.7 | 27.4 | 26.2 | 26.7 |
| 0-shot | FOCUS | 69.2 | 63.8 | 45.7 | 60.4 | 38.8 | 55.6 | 21.9 | 1.8 | 0.0 | 20.1 | 22.6 | 13.3 |
| | ours | **79.7** | **77.5** | **73.0** | **81.9** | **53.0** | **73.0** | **23.8** | **17.7** | **18.9** | **28.7** | **26.8** | **23.2** |
| $n$-shot | FOCUS@800M | 74.8 | 74.3 | 72.4 | 73.3 | 48.9 | 68.7 | 24.4 | 17.1 | 22.6 | 22.6 | 26.2 | 22.6 |
| | ours@800M | **80.9** | **80.7** | **77.8** | **80.7** | **54.4** | **74.9** | **28.0** | **25.0** | **26.2** | **29.9** | **28.7** | **27.6** |

# 4 Experiments

## 4.1 Setup

**Data.** We use the English subset of the MADLAD-400 corpus (Kudugunta et al., 2023) and code from the StarCoder data (Li et al., 2023) for hypernetwork training. The sampling ratio of English to Code is 7:3 following Zhang et al. (2024). For the multilingual hypernetwork, we use a subset of 26 of the languages used in XGLM (Lin et al., 2022).[6] with data from MADLAD-400. We sample languages using a multinomial distribution as in Conneau & Lample (2019) with $\alpha = 0.1$. For the $n$-shot experiments, we also train on the StarCoder data, but substitute the English section of the MADLAD-400 corpus for Flan v2 (Longpre et al., 2023) sampled as in Soldaini et al. (2024).[7]

**Evaluation.** We use the standard benchmarks PiQA (Bisk et al., 2020), HellaSwag (HS; Zellers et al., 2019), BoolQ (Clark et al., 2019), MMLU (Hendrycks et al., 2021) and the "easy" subset of ARC (Clark et al., 2018) for evaluation in English and the synthesis task of HumanEvalPack (Muennighoff et al., 2023) for coding evaluation. For multilingual evaluation, we use XNLI (Conneau et al., 2018), XCOPA (Ponti et al., 2020) and MMLU as machine-translated by Lai et al. (2023).

---

[6]We exclude languages without whitespace between words since they would require language-specific pretokenizers (e.g. Sun, 2012). Although our method is also applicable to this case, we leave this to future work.

[7]We use Flan v2 because we observed a strong decrease in accuracy from continuing to train on the MADLAD-400 data (even with the original tokenizer). The training data for most LLMs (including Mistral-7B) is not public, but it is plausible that this decrease stems from higher-quality data mixed in especially towards the end of training as in e.g. Groeneveld et al. (2024).

Table 3: Accuracy of Mistral-7B on XCOPA with language-specific tokenizers zero-shot transferred via FOCUS and our hypernetwork. The standard errors are between 2.1% and 2.3%.

| | et | ht | id | it | qu | sw | ta | tr | vi | Avg. |
|---|---|---|---|---|---|---|---|---|---|---|
| original | 46.6 | 51.6 | 58.0 | 65.8 | 48.4 | 51.4 | 54.4 | 56.4 | 59.0 | 54.6 |
| FOCUS | 52.0 | 53.0 | 51.2 | 49.2 | **51.4** | 54.6 | 54.0 | 55.2 | 49.8 | 52.3 |
| ours | **53.4** | **57.2** | **60.0** | **65.6** | 50.0 | **57.2** | **55.8** | **57.4** | **57.2** | **57.1** |
| $\Delta$accuracy | +7% | +6% | +2% | 0% | +1% | +6% | +1% | +1% | -2% | +3% |
| $\Delta$length | -72% | -42% | -52% | -36% | -54% | -51% | -83% | -57% | -59% | -54% |

Table 4: 5-shot accuracy of Mistral-7B on multilingual MMLU with the original tokenizer and language-specific tokenizers zero-shot transferred via FOCUS and our hypernetwork.

| | original | FOCUS | ours | $\Delta$accuracy | $\Delta$length |
|---|---|---|---|---|---|
| German | 51.6 | 26.2 | **43.7** | -8% | -37% |
| Spanish | 53.6 | 26.2 | **45.9** | -8% | -32% |
| French | 53.6 | 27.4 | **44.8** | -9% | -30% |
| Italian | 52.5 | 25.8 | **42.7** | -10% | -36% |
| Russian | 49.9 | 27.2 | **35.1** | -15% | -47% |

**Models.** To evaluate our method, we use Mistral-7B (Jiang et al., 2023) as the main decoder-style language model and XLM-R (Conneau et al., 2020) as a representative of encoder-style models.[8] We also experiment with the smaller TinyLlama-1.1B model (Zhang et al., 2024) in Appendix H.

**Tokenizers.** We transfer models to the GPT2 tokenizer (Radford et al., 2019) for evaluation on natural language benchmarks and to the StarCoder tokenizer (Li et al., 2023) for evaluation on code benchmarks.[9] For multilingual evaluation, we train language-specific monolingual tokenizers with a vocabulary size of 50k using SentencePiece (Kudo & Richardson, 2018) and evaluate transfer to these. We also verify that the hypernetwork is robust to the choice of vocabulary size in Appendix E.

**Hypernetwork training.** We train the hypernetwork for 200k steps (10k of which are MIMICK-style warmup) with a batch size of 128 and a sequence length of 128 (we find it sufficient to use short sequence lengths).[10] For the multilingual decoder-style models, we start from the English + Code checkpoint and forgo MIMICK-style warmup, keeping other hyperparameters unchanged. We use a RoBERTa-style architecture i.e. bidirectional attention and Post-LayerNorm Transformer layers (Liu et al., 2019), but use a feedforward dimension of 2x the hidden dimension instead of 4x for the hypernetwork. See Appendix D for a full list of hyperparameters.

**Continued training details.** To keep runtime comparable between training the model with hypernetwork and direct training (without hypernetwork), we run hypernetwork inference only for a subset of $k = 16384$ tokens in the continued training case. The subset consists of all tokens occurring in the batch, plus a uniform sample of those that do not occur. The language modeling loss is then only computed over this subset of tokens. We found in preliminary experiments that this causes only minor performance degradation. Furthermore, we use the zero-shot predicted embeddings as the target for the auxiliary loss instead of using the original embeddings. This stabilizes training. We train for 50k steps with a batch size of 32 and sequence length of 512, resulting in 'seeing' 819.2M tokens.

### 4.2 Zero-Shot and n-shot Results

Results for XLM-R are shown in Table 1. We take task adapters trained for the original XLM-R model on the English XNLI dataset via Poth et al. (2023) and substitute the tokenizer for our language-specific one. We compare our hypernetwork against a simple lexical baseline (copying the

---

[8]Although (decoder-style) LLMs are the centerpiece of a large amount of current NLP research, encoder-style LMs have wide-ranging applications in e.g. retrieval (Khattab & Zaharia, 2020) and LLM distillation (Hsieh et al., 2023) due to their lower computational cost.

[9]We chose these tokenizers due to their popularity and comparatively efficient encoding of the target domain.

[10]Training takes around one day for the XLM-R hypernetwork on a TPU v3-8 and three days for the Mistral-7B hypernetwork on a TPU v4-32 pod.

Table 5: Single model rating results on MT-Bench of transferring Mistral-7B-Instruct-v0.1 to the GPT2 tokenizer using the hypernetwork trained for the base Mistral-7B model. We use `gpt-3.5-turbo-1106` as a judge. *orig.* is the original fine-tuned model, *base* the model with the same tokenizer but embeddings substituted for the base models' embeddings. $\lambda$ is the scaling factor for the weight differences in Task Arithmetic (Ilharco et al., 2023).

| | original | | 0-shot | | n-shot | | | |
|---|---|---|---|---|---|---|---|---|
| Embeddings | orig. | base | FOCUS | ours | ours@800 | | | |
| $\lambda$ | - | - | - | - | 0.0 | 0.3 | 0.5 | 0.7 |
| Score (1 to 10) | 7.33 | 7.48 | 5.03 | **6.56** | 6.59 | 6.75 | **6.82** | 6.77 |

embeddings of overlapping tokens and initializing the rest randomly), FVT, OFA, and FOCUS (c.f. Section 2). We focus only on FOCUS in the following since it performs best among the baselines. Our hypernetwork consistently outperforms all baselines and preserves accuracy to 1% on average, losing 3% in the worst case and improving by 1% in the best case, while sequences are on average 14% shorter for the language-specific tokenizers; inference is thus more than 16% faster.[11] We show in Appendix E that these results are robust to the target vocabulary size.

Table 2 shows results on English and Code for Mistral-7B. We find that ZeTT is more challenging in the decoder case: FOCUS performs roughly random in the worst case (-23.2% on BoolQ) and is reduced to 0% pass@1 on HumanEval in Python. The hypernetwork goes a long way in closing this gap but still falls behind on some benchmarks. However, continuing to train the hypernetwork with the target tokenizer closes the gap almost completely. In fact, continued training on 800M tokens with the StarCoder tokenizer performs *better* than continued training for the same amount of tokens with the original tokenizer, potentially because the StarCoder tokenizer is more well suited towards code; it results in approx. 10% less tokens on average. Also, notably, continued training with the original tokenizer slightly *degrades* performance on average; this may be due to a higher-quality data mix used for pretraining Mistral-7B, whereas we use public data sources (c.f. Section 4.1).

Results of the multilingual hypernetwork for Mistral-7B are shown in Table 3 and Table 4. On XCOPA, the hypernetwork on average improves performance over the original model, while also more than halving sequence length. XCOPA performance is close to random in some languages (e.g. Southern Quechua (qu) and Estonian (et)), so we also evaluate on multilingual MMLU. Here, although the hypernetwork clearly outperforms FOCUS (which performs close to random), there is still a substantial gap to the original model; this could presumably be fixed via continued training.

### 4.3 Applying a Hypernetwork trained for a Base Model to Fine-Tuned Models

A large amount of the models used by practitioners are fine-tuned versions of base models[12], e.g. via SFT or RLHF (Ouyang et al., 2022). We now attempt to answer the question: *Given a hypernetwork trained for a base model, can we apply this hypernetwork to fine-tuned versions of the same model without any extra training?* This would act as a multiplying factor for the hypernetwork's applicability. First, we observe that the embedding space of a fine-tuned model is compatible with that of the base model: the embeddings of the fine-tuned Mistral-7B-Instruct-v0.1 have an average cosine similarity of 98.6% to the corresponding embedding in the base model while the average cosine similarity of the mean embedding vector is 17.4%.[13] Embedding compatibility also holds true for other models (Appendix H). The predictions of a hypernetwork trained for a base model can thus be used out-of-the-box with fine-tuned models. We verify that this is the case by evaluating Mistral-7B-Instruct-v0.1 transferred to the GPT2 tokenizer on the corrected[14] version of MT-Bench (Zheng et al., 2023). For $n$-shot transfer, since we train the full model we also need a way to transfer the non-embedding parameters; we achieve this via Task Arithmetic (Ilharco et al., 2023). Results are shown in Table 5. The transferred fine-tuned model performs well, coming within approx. 0.5 score of the original model. Also, curiously, the fine-tuned model with the original tokenizer performs *better* when using the embeddings of the (not fine-tuned) base model; this may be a prudent direction for future work.

---

[11]1/(1-14%)=16%, plus additional speedup due to attention scaling quadratically with sequence length.

[12]We refer to models purely pretrained on the Language Modeling task as *base models*.

[13]Averaged across the input and the output embeddings.

[14]Using the corrections from `https://github.com/InflectionAI/Inflection-Benchmarks`.

Table 6: NLI performance on Farsi (FarsTail; Amirkhani et al., 2023), Dutch (SICK-NL; Wijnholds & Moortgat, 2021), Aymara and Guarani (AmericasNLI; Ebrahimi et al., 2022). We measure zero-shot transfer from a model trained on English XNLI (c.f. Table 1), except for Sick-NL where we train an adapter on SICK (Marelli et al., 2014) since the XNLI adapter underperforms.

| | Unseen by Hypernet | | Completely Unseen | |
| | **Farsi** | **Dutch** | **Aymara** | **Guarani** |
|---|---|---|---|---|
| original | 72.4 | 76.6 | 40.0 | 42.4 |
| Lexical | 60.5 | 72.7 | 38.5 | 41.7 |
| FVT | 65.5 | 74.8 | 38.4 | 39.0 |
| FOCUS | 65.3 | 74.8 | 37.7 | 40.7 |
| ours | **66.4** | **77.8** | **42.9** | **42.2** |
| $\Delta$accuracy | -6% | +1% | +3% | 0% |
| $\Delta$length | -12% | -19% | -36% | -39% |

Table 7: Performance of Mistral-7B transferred to the GPT2 tokenizer on English benchmarks (c.f. Table 2), as well as transferred to a tokenizer containing all words in the evaluation datasets; this converts Mistral-7B to a *word-level* language model on the evaluation corpora.

| | | PiQA | HS | ARC | BoolQ | MMLU | Avg. |
|---|---|---|---|---|---|---|---|
| original | | 80.7 | 81.0 | 79.5 | 83.6 | 59.6 | 76.9 |
| GPT2 Tokenizer | FOCUS | 69.2 | 63.8 | 45.7 | 60.4 | 38.8 | 55.6 |
| | ours | **79.7** | **77.5** | **73.0** | **81.9** | **53.0** | **73.0** |
| | $\Delta$length | -7.8% | -5.6% | -6.1% | -13.1% | -9.9% | -8.5% |
| Word Tokenizer | FOCUS | 66.8 | 58.8 | 51.3 | 62.6 | 35.2 | 54.9 |
| | ours | **78.9** | **74.9** | **73.9** | **80.9** | **49.4** | **71.6** |
| | $\Delta$length | -14.6% | -10.1% | -14.9% | -20.3% | -16.8% | -15.3% |

# 5 Discussion

**Converting tokenizers to byte-level.** As per Section 3.2, we need a procedure to convert tokenizers to the byte level to ensure that token decomposition is always possible. This is trivial in most cases; the missing bytes just need to be added to the vocabulary. BPE is an exception: here, we need to change the units on which merges are defined from characters to bytes. We achieve this by adding merges to assemble the characters used by the tokenizer from their constituent bytes to the beginning of the merge table. This preserves the tokenization in more than 99% of cases (Appendix J).

**Converting tokenizers to UnigramLM.** We also introduce a procedure to convert arbitrary tokenizers to tokenizers using UnigramLM as the tokenization function. We refer to this process as *unigramifying* (details in Appendix A). An important assumption of the hypernetwork training is that by using the UnigramLM parametrization with scores distributed as Gaussians we can cover a sufficiently diverse distribution of tokenizers for the hypernetwork to generalize to e.g. BPE tokenizers. Unigramifying allows us to check if, in principle, this is possible. Luckily, we find that it is: unigramifying results in minimal performance degradation when substituting the original tokenizer with the corresponding UnigramLM tokenizer (Appendix J). Although this does not guarantee that our distribution of tokenizers is sufficiently diverse, our empirical results suggest it is (cf. Section 4.2).

We believe our conversion methods to UnigramLM and to byte-level will simplify further research into tokenizer transfer, showing that *the wildly heterogeneous landscape of tokenizers can be well approximated via byte-level UnigramLM tokenizers*.

**What is the effect of amortizing over the tokenization function?** As described earlier in Section 3, we 'amortize' over the tokenization function, that is, the tokenization function is not an input to our hypernetwork. We find that the predicted amortized embeddings are robust to the choice of tokenization function. For example, the set of embeddings predicted for the GPT2 vocabulary has low bits-per-character for both the original GPT2 tokenization function and a different UnigramLM tokenization function with scores based on token frequencies (Appendix J). This is not the case

for the original GPT2 embeddings: while they (as expected) perform well with the original GPT2 tokenizer, there is significant performance degradation when switching to the frequency-based UnigramLM tokenization function. This calls into question prior work copying the embeddings of overlapping tokens for transfer across tokenizers (Dobler & de Melo, 2023; Gee et al., 2022, among others), indicating that *even if there is an exactly overlapping token in the original tokenizer, it is not necessarily the optimal initialization of the corresponding token in the new tokenizer*.

Although we amortize over most of the aspects of the tokenization function, in practice, tokenization functions rely on a considerable amount of engineering, so it is not possible to amortize over everything; we discuss remaining assumptions in Appendix I.

**Analyzing computational overhead.** We estimate the FLOPs per token of multiple hypernetworks in Appendix K. Given a batch size $n$ and sequence length $s$ for the main model, and using the hypernetwork to compose $k$ token sequences of length $t$, the FLOPs per batch will be $n \times s \times (\frac{\text{FLOPs}}{\text{token}})_{\text{main}} + k \times t \times (\frac{\text{FLOPs}}{\text{token}})_{\text{hypernet}}$. Taking Mistral-7B as an example with $n = s = 128$, $k = 32768$ and $t = 7$ the FLOPs per batch will be 252T + 30T i.e. a 12% overhead from applying the hypernet. Notably, we observed that a hypernetwork size of three layers is sufficient, regardless of the main model, so the relative overhead decreases with increased amounts of layers in the main model.

**Generalization to unseen tokens.** Although our primary goal is generalization to unseen *tokenizers* (i.e., tuples $(\mathcal{V}, T)$), the question of how well our hypernetwork can generalize to unseen *tokens* (elements of $\mathcal{V}$) presents itself. To answer this question, we test the XLM-R and Mistral-7B hypernetworks on out-of-distribution vocabularies. Specifically, we test the XLM-R hypernetwork on Farsi and Dutch (which are unseen by the hypernet, but seen by the base model) as well as Aymara and Guarani, which are unseen by both. Table 6 confirms the hypernet performs well in this case, even gaining in performance over the model with original embeddings in completely unseen languages. In this setup, up to 40% of the used tokens in the target vocabularies have never been seen during hypernetwork training (we analyze this overlap in detail in Appendix G). The reason for the performance increase from the hypernetwork on unseen languages may be that, under the original tokenization, the embeddings of many tokens occuring in unseen languages are undertrained (c.f. Land & Bartolo, 2024), while the embeddings produced by the hypernetwork do not suffer from this issue; future work could investigate this in more detail. For Mistral-7B, we instead transfer to an out-of-distribution *word-level* tokenizer by creating a tokenizer which contains all words which occur in any evaluation corpus (approx. 100k in total). 3.3k words are completely unseen and 13.5k words have been seen in less than 0.1% of training steps. Still, performance only deteriorates by a small amount and the improvement over FOCUS persists as shown in Table 7.

## 6 Conclusion

We have established *Zero-Shot Tokenizer Transfer (ZeTT)*, the difficult problem of transferring language models to a new tokenizer without any training. We have found that prior heuristics for embedding initialization provide a first baseline for ZeTT, but fall short in many cases. To establish a much stronger baseline, we introduced a hypernetwork-based approach that closes the gap to a large extent, and can be further improved via continued training on a few (<1B) tokens. Due to preserving the embedding space of the original model, ZeTT can be applied to e.g. reusing adapters trained for the original model with a different tokenizer, and to transferring fine-tuned models to a new tokenizer using a hypernetwork trained for the base model. In aggregate, this work is a substantial step towards *detaching* language models from their tokenizer, increasing their flexibility and reusability.

## 7 Limitations

The key limitation of our approach is the requirement to train a hypernetwork for every base model. Although the hypernetwork only needs to be trained once, doing so is computationally intensive and may not be feasible for many LLM practitioners. Instead, it may be a task LLM providers are better positioned to undertake. Other limitations are the remaining assumptions on the tokenization function (Appendix I), and not taking the tokenization function into account (Appendix J), although these limitations do not appear to have substantial impact in practice. Finally, we have limited our scope to experiments on text-only models, but Zero-Shot Tokenizer Transfer could also be beneficial for multimodal models, such as models 'perceiving' images or speech; we leave this to future work.

# Acknowledgments

This work has been supported by a Royal Society University Research Fellowship *'Inclusive and Sustainable Language Technology for a Truly Multilingual World'* (no 221137; 2022-) awarded to Ivan Vulić. Research supported with Cloud TPUs from Google's TPU Research Cloud (TRC). We thank Markus Frohmann, Marcell Fekete and Piotr Nawrot for helpful feedback on a draft of this paper, and Arduin Findeis for many valuable discussions during the entirety of this project.

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

# A   Unigramifying: Approximating Arbitrary Tokenizers via UnigramLM

We introduce a procedure to convert arbitrary tokenizers to UnigramLM in an optimal (but lossy) way which we refer to as *unigramifying*. Given a text $x$ and the sequence of tokens $T(x)$, for the UnigramLM tokenizer $\hat{T}$ to be equivalent to $T$, it is necessary that $\hat{T}$ fulfills $\sum_{t \in T(x)} \log p_{\hat{T}}(t) > \sum_{t \in C} \log p_{\hat{T}}(t)$ for all $C$ in $\mathcal{C}_x \setminus \{T(x)\}$.[15] Thus, given a corpus of texts $X$ we can formulate a loss

$$\mathcal{L}_\mathcal{T}(X, \hat{T}) = \sum_{x \in X} \sum_{C \in \mathcal{C}_x \setminus \{T(x)\}} \max\left(0, \sum_{t \in C} \log p_{\hat{T}}(t) - \sum_{t \in T(x)} \log p_{\hat{T}}(t)\right)$$

which is zero if and only if the condition above is satisfied for all texts in $X$. This objective is piecewise linear, so it can be converted to a standard Linear Programming (LP) form and solved via an LP solver. In practice, we use the CPLEX v22.1 (IBM ILOG, 2022) solver. Since applying the procedure to a corpus directly would be costly, we first pre-tokenize the training corpus, then count the pretokens, and choose the top $n = 1000000$ pretokens as the set $X$.

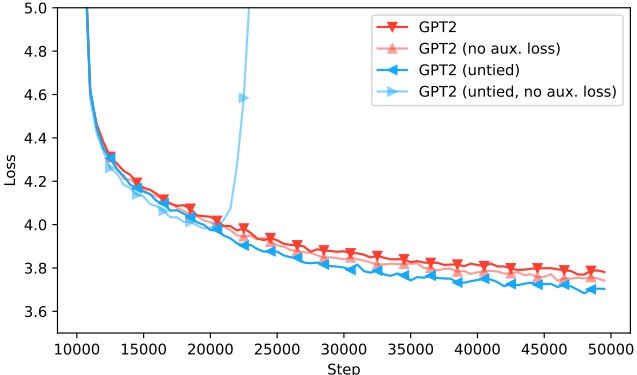

Figure 3: Language modeling loss of GPT2, and GPT2 with untied weight embeddings with and without the auxiliary loss across the first 50k training steps, excluding MIMICK-style warmup.

# B   Stabilization Effect of the Auxiliary Loss

We found in preliminary experiments that the auxiliary loss is necessary, especially for models that do not share embedding parameters between the input and the output (models with *untied* embeddings). To validate this hypothesis, we conducted an experiment where we manually untied the embeddings of GPT2 i.e. used a separate hypernetwork prediction head for the input and the output embeddings. Although everything else is kept the same, the untied GPT2 model diverges without the auxiliary loss, whereas the original GPT2 trains as expected, even without an auxiliary loss (Figure 3).

# C   Non-Amortizing Hypernetworks

We experimented with hypernetworks taking the tokenization function into account by adding *sparse inter-token attention* blocks between the self-attention and the FFN in every hypernetwork layer. Sparse inter-token attention consists of two attention blocks. The first attention block attends from a fixed amount of learnable inter-token embeddings (e.g. 16, each a vector of size $d_{\text{model}}$) to the $i$th token representation of every token sequence passed to the hypernetwork. The second block attends from the $i$th token representation to the inter-token embeddings. This way, we factorize the attention to e.g. one $16 \times k$ attention and one $k \times 16$ attention, instead of the standard $k \times k$ self-attention

---

[15]This is not sufficient for equivalence since order is ignored e.g. $T(x) = \{ab, a, b\}$ and $\hat{T}(x) = \{a, b, ab\}$ fulfill the criterion but are not equivalent.

Table 8: Performance of the hypernetwork in bits-per-byte with and without inter-token attention. *Sampled Tokenizers* are tokenizers as sampled during the training loop (c.f. Algorithm 1), *en* is an English UnigramLM tokenizer. The respective vocabulary sizes are shown in brackets.

|                              | Sampled Tokenizers (32k) | GPT-NeoX (50k) | en (30k) |
| ---------------------------- | ------------------------ | -------------- | -------- |
| ours                         | 1.157                    | **0.902**      | **1.054** |
| ours (+ inter-token attention) | **1.118**              | 0.904          | 1.103    |

which would be infeasibly slow for typical vocabulary sizes. We only add inter-token attention for the first token in every sequence. This improves performance on the sampled tokenizers, but does not improve performance on 'real-world' tokenizers (Table 8); investigating this mismatch is a direction for future work.

# D    Additional Hyperparameters

Hyperparameters for hypernetwork training are shown in Table 9. For continued training, we use the same optimizer, but a sequence length of 512, batch size of 32, training for 50k steps and a constant learning rate chosen among the set $\{1e{-}6, 3e{-}6, 6e{-}6, 1e{-}5, 3e{-}5\}$ to maximize performance. The chosen learning rate is $1e{-}6$ for the runs keeping the original tokenizer (*original@800M*), $6e{-}6$ for continued training starting from FOCUS (*FOCUS@800M*) and $3e{-}6$ for continued training with the hypernetwork (*ours@800M*).

Table 9: Hypernetwork hyperparameters.

| | |
| --- | ---: |
| Optimizer | AdamW (Loshchilov & Hutter, 2019) |
| $(\beta_1, \beta_2)$ | (0.9, 0.95) |
| weight decay | 0.01 |
| Max. global gradient norm | 0.1 |
| Sequence length | 128 |
| Batch size | 128 |
| Steps | 200000 |
| of which MIMICK-style warmup steps | 10000 |
| MIMICK-style warmup learning rate schedule | linear warmup to 3-e4 |
| Main learning rate schedule | linear warmup to 6e-5 until 10k, then cosine decay to 6e-6 |
| Tokenizer sampling | |
| Vocabulary size | 32768 |
| Distribution of noise level $z$ | $\mu = \ln(10^{-5}), \sigma = 4$ |
| Batch size $m$ | 2048 |
| Auxiliary loss weight | 0.5 |
| Hypernetwork | |
| num. layers | 3 |
| max. sequence length | 7 (English + Code) or 15 (multilingual) |
| hidden dimension | $d_{\text{model}}$ |
| FFN dimension | $2d_{\text{model}}$ |
| num. attention heads | $\min(d_{\text{model}}/64, 32)$ |

# E    Sensitivity to Tokenizer Size

Since the tokenizers we experiment with have similar vocabulary sizes (50k for the language-specific tokenizers and for GPT2, 49k for the StarCoder tokenizer) we conduct an additional experiment to quantify the sensitivity of the performance of our hypernetwork to the size of the target tokenizer. We find that although there is slight performance degradation when increasing the size of the new tokenizers' vocabulary, the hypernetwork is fairly robust to vocabulary size (Figure 4).

# F    Reliance on Vocabulary Overlap

Intuitively, transfer is easier the more the target has in common with the source. One way to measure commonality between the original (source) and the target tokenizer is the fraction of tokens of the

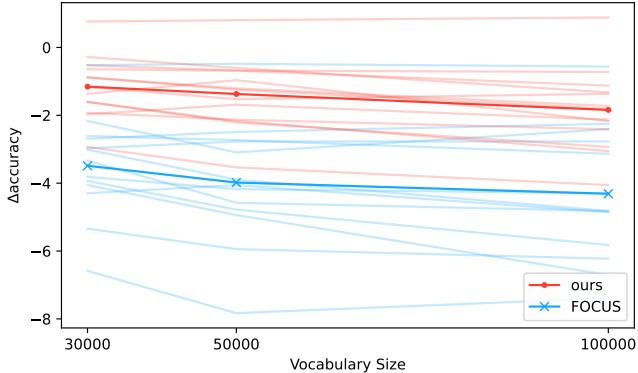

Figure 4: Difference in accuracy to the original XLM-R model on XNLI of our method and FOCUS across vocabularies with size 30k, 50k, and 100k of the new tokenizer.

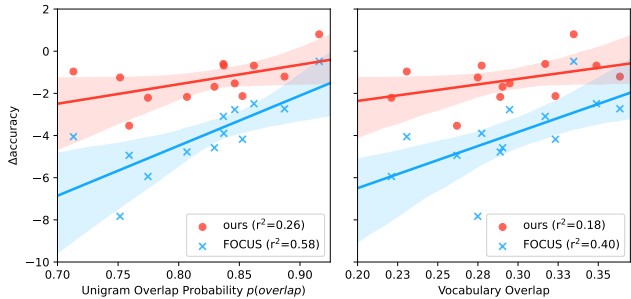

Figure 5: Correlation of the difference in accuracy to the original XLM-R model with Unigram overlap probability $p(\text{overlap})$ (left) and vocabulary overlap (right).

target vocabulary which also exist in the source vocabulary (*vocabulary overlap*). Performance correlates with vocabulary overlap, but it correlates more strongly with the probability for tokens to overlap: that is, when randomly sampling some token from a corpus tokenized with $T_b$, the probability that this token also exists in the vocabulary of $T_a$. We refer to this metric as $p(\text{overlap})$. $p(\text{overlap})$ has higher correlation with the performance of FOCUS, indicating that our hypernetwork depends less on overlap (Figure 5).

Table 10: Performance of TinyLlama-1.1B after zero-shot and $n$-shot tokenizer transfer (training on 800M tokens), compare Table 2.

| #shots | Method | Natural Language ($\rightarrow$ GPT2 Tok.) | | | | | | Code (pass@1) ($\rightarrow$ StarCoder Tok.) | | | | | |
|---|---|---|---|---|---|---|---|---|---|---|---|---|---|
| | | | | | | | | | HumanEvalPack | | | | |
| | | PiQA | HS | ARC | BoolQ | MMLU | Avg. | js | go | py | cpp | java | Avg. |
| original | | 73.1 | 59.1 | 55.2 | 57.2 | 25.5 | 54.0 | 7.3 | 6.7 | 7.3 | 8.5 | 7.9 | 7.5 |
| original@800M | | 73.2 | 59.5 | 63.3 | 65.1 | 26.3 | 57.5 | 9.8 | 7.3 | 9.1 | 8.5 | 10.4 | 9.0 |
| 0-shot | FOCUS | 60.8 | 42.1 | 39.6 | 56.9 | 22.9 | 44.7 | 4.9 | 0.6 | 0.0 | 3.0 | **7.9** | 3.3 |
| | ours | **70.5** | **55.6** | **51.4** | **62.9** | **23.7** | **52.8** | 4.3 | **5.5** | **4.3** | **7.3** | 3.7 | **5.0** |
| $n$-shot | FOCUS@800M | 67.7 | 52.8 | 52.7 | **66.1** | 25.3 | 52.9 | 6.1 | **6.1** | 10.4 | 8.5 | **8.5** | 7.9 |
| | ours@800M | **71.4** | **57.8** | **59.7** | **66.1** | **26.6** | **56.3** | **9.1** | **6.1** | **11.6** | **11.0** | 7.3 | **9.0** |

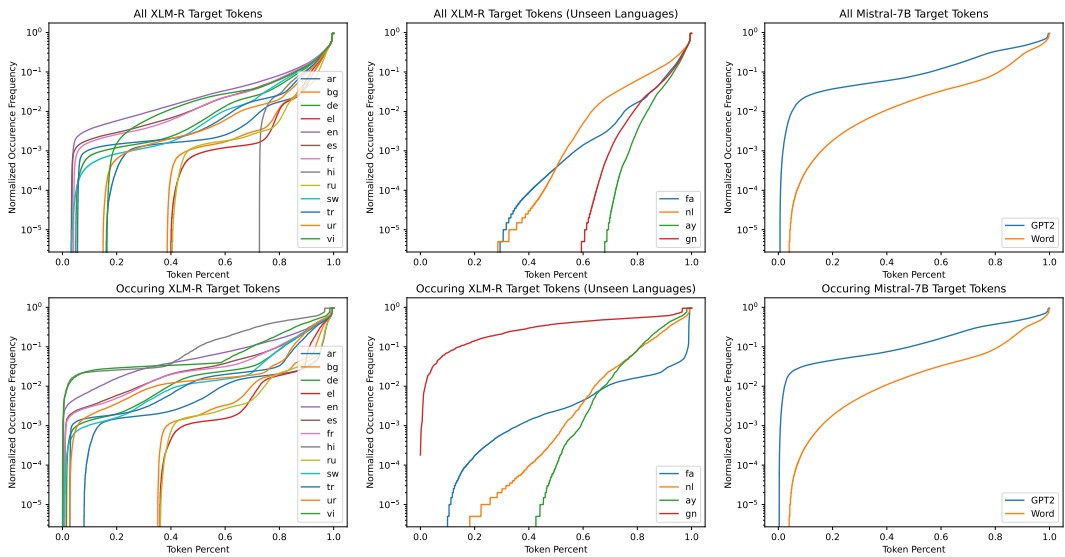

Figure 6: Analyzing how often the hypernetwork sees the tokens of different target tokenizers during training. Note the logarithmic y-scale. We analyze the occurrence for all tokens in the target vocabulary (top) and for tokens which occur at least once in the evaluation data (bottom) across target tokenizers in seen languages for XLM-R (left), unseen XLM-R languages (middle) and English Mistral-7B tokenizers (right). The bottom row is more informative w.r.t. how well the hypernetwork generalizes to unseen tokens since tokens which do not occur do not substantially impact evaluation.

Table 11: Single model rating results on MT-Bench of transferring TinyLlama-1.1B-Chat-v1.0 to the GPT2 tokenizer, compare Table 11.

|  | original | | 0-shot | | n-shot | | | |
|---|---|---|---|---|---|---|---|---|
| Embeddings | orig. | base | FOCUS | ours | ours@800 | | | |
| $\lambda$ | - | - | - | - | 0.0 | 0.3 | 0.5 | 0.7 |
| Score (1 to 10) | 5.5 | 5.7 | 2.7 | **4.0** | 4.29 | 4.63 | **4.8** | 4.43 |

## G Reliance on Overlap between Hypernet Training Tokens and Target Tokens

We analyze how often the hypernetwork sees the tokens in the vocabulary of different target tokenizers across multiple settings in Figure 6. We differentiate between tokens which occur in the evaluation data, and tokens which do not; this is important since the embeddings of tokens which do not occur in the evaluation data will not substantially impact performance. Notably, for XLM-R, >35% of occurring tokens in Greek, Bulgarian and Russian are unseen by the hypernet, even though the hypernet is trained on these languages. This is likely due to the non-Latin scripts. The hypernet still performs well in these languages with an average 2% performance decrease at 17% sequence length reduction on XNLI. In total, the HN has seen approx. 200M different tokens during training.

## H Additional LLM Results

Zero-shot and n-shot results for TinyLlama-1.1B are shown in Table 10 and MT-Bench results of transferring TinyLlama-1.1B-Chat-v1.0 in Table 11. We observe the same patterns as on Mistral-7B.

## I Assumptions on the Tokenization Function

In practice, besides the tokenization algorithm itself (e.g. BPE, UnigramLM) tokenization functions also contain other steps, in particular *pretokenizing* text into smaller chunks (usually words) on which to apply the tokenization function (Mielke et al., 2021). In our experiments, we assume fixed

Table 12: Probability of pretokens sampled from the English MADLAD-400 data to be tokenized equivalently to the original tokenization when converting the tokenizer to byte-level (*To Byte-Level*) or to UnigramLM (*Unigramify*). Also shown is the LMs bits-per-character when applying the original vs. the corresponding UnigramLM tokenizer. Bits-per-character can not be measured for conversion to byte-level since extra tokens are added in this process (which there are no embeddings for).

| Kind | | **BERT** WordPiece | **Mistral-7B** BPE | **TinyLlama-1.1B** BPE | **GPT2** BBPE |
|---|---|---|---|---|---|
| Original | $p$(preserved) | 100% | 100% | 100% | 100% |
| | bits per char | n/a | 0.675 | 0.747 | 0.930 |
| To Byte-Level | $p$(preserved) | 99.6% | 99.9% | 99.9% | 100% |
| | Extra Tokens | 162 | 522 | 362 | 0 |
| Unigramify | $p$(preserved) | 99.4% | 99.8% | 99.8% | 99.7% |
| | bits per char | n/a | 0.678 | 0.750 | 0.932 |

Table 13: Bits-per-character of GPT2 with the original tokenizer and the tokenization function being original (left), unigramified (middle) and UnigramLM with scores set to the substring frequency of the tokens (right). We compare the original embeddings with embeddings predicted from our hypernetwork, with or without Gaussian noise in the sampling process.

| Model | Embeddings | Tokenizer $(\mathcal{V}, T)$ | | |
|---|---|---|---|---|
| | | $(\text{GPT2}, \text{GPT2})$ | $(\text{GPT2}, \text{unigramify}(\text{GPT2}))$ | $(\text{GPT2}, \text{UnigramLM})$ |
| GPT2 | original | 0.930 | 0.932 | 1.005 |
| | ours | **0.919** | **0.920** | **0.964** |
| | ours (no noise) | 0.925 | 0.926 | 0.978 |

pretokenization given by a regular expression based on the regular expression used by GPT2 (Radford et al., 2019), adjusted to not over-segment text in languages using characters in the Unicode Mark category within words (e.g. Hindi and Tamil). We also add a *prefix space* (i.e., a whitespace at the start of the text to tokenize) if and only if the original tokenizer also uses a prefix space. Finally, we always add whitespace characters covering sequences of consecutive whitespaces up to 16 characters long similar to Black et al. (2022) to ensure code is tokenized efficiently. These light assumptions mostly preserve the generality of our method but could be further relaxed in future work.

## J    Tokenization Function Amortization and Unigramifying Results

Results measuring the success of unigramifying tokenizers are shown in Table 12. Results measuring the success of amortizing over the tokenization function are shown in Table 13.

Table 14: Parameter count and FLOPs estimates for our hypernetwork (and the corresponding main model) in different setups. The relatively lower computational cost compared to parameter count is mainly due to forgoing de-embedding which contributes significantly to FLOPs (Kaplan et al., 2020).

| | **Model** | | **Hypernet** | |
|---|---|---|---|---|
| | #params | FLOPs / token | #params | FLOPs / token |
| GPT2 | 124M | 253M | 21M (16%) | 4.5M (1.8%) |
| TinyLlama-1.1B | 1.1B | 2.1G | 170M (15%) | 33.1M (1.6%) |
| Mistral-7B | 7.2G | 15.4G | 678M (9%) | 132.1M (0.9%) |

## K    Analyzing FLOPs of the hypernetwork

Estimated FLOPs per token for the hypernet and the corresponding main model are shown in Table 14. We estimate FLOPs on the basis of XLA-compiled instructions using Jax (Bradbury et al., 2018).

