# OpenReview forum: "Zero-Shot Tokenizer Transfer"
_NeurIPS.cc/2024/Conference — NeurIPS 2024 poster_

### Official Review · Reviewer_1Y74 · 2024-07-03

**Soundness:** 4
**Presentation:** 4
**Contribution:** 3
**Rating:** 7
**Confidence:** 4

**Summary:**

The paper presents a novel approach towards separating the tokenizer from the language model (LM). All modern LMs are trained with a fixed tokenizer, which prevents them from generalizing well to unseen or rarely seen tokens. Furthermore, the bounding to a specific tokenizer prevents models trained with different tokenizers from being merged and ensembled. To overcome these issues, the authors propose training a hypernetwork capable of mapping embedding parameters between tokenizers. The authors perform extensive experiments to confirm the effectiveness of the proposed approach, even when the tokenizer is transferred in a zero-shot setting.

**Strengths:**

The paper is well-written and easy to follow. The authors describe a zero-shot tokenizer transfer problem and propose a state-of-the-art solution for transferring language models to a new tokenizer without additional training. Unlike popular heuristic-based approaches, the authors propose a novel approach that involves training a hypernetwork. Especially impressive is the attention to the technical details described in the paper, for example, the text sampling strategy or the design of the appropriate loss function. Furthermore, the authors experiment with models of different architectures, different tokenizers, and on a variety of tasks, and verify the effectiveness of the proposed approach.

**Weaknesses:**

In my opinion, the paper is technically sound and contains an appropriate amount of explanation and evaluation. If anything, I would like to see two additional aspects explored: first, more experiments demonstrating how this approach would scale with increasingly large language models (LLMs) in question; and second, experiments with other open-sourced LLMs.

**Questions:**

1. Experiments with TinyLlama show a similar trend to that of the Mistral results. However, I think it's important for the reader to see that the approach works with different models; I would recommend moving Appendix G to the main text.

2. You performed experiments with small language models, up to 7 billion parameters. How do you think your approach would generalize to models with a higher number of parameters, and why?

3. Do you think a similar approach would work for other modalities, such as multimodal models?

**Limitations:**

Yes

---

> ### Author Rebuttal · Authors · 2024-08-06
>
> Thank you for your feedback and ideas toward extending our approach!
>
> > Experiments with TinyLlama show a similar trend to that of the Mistral results. However, I think it's important for the reader to see that the approach works with different models; I would recommend moving Appendix G to the main text.
>
> Thanks, in case of acceptance we will use the additional page to move the TinyLlama results to the main paper. We are also working on training hypernetworks for more recent LLMs such as Llama 3.1, we will add these results along with the TinyLlama results once the experiments are finished.
>
> > You performed experiments with small language models, up to 7 billion parameters. How do you think your approach would generalize to models with a higher number of parameters, and why?
>
> Our evidence suggests our approach scales favorably with parameter count since the amount of layers in the hypernetwork can mostly stay constant while the amount of layers in the base model increases (Appendix J). The relative overhead of applying the hypernetwork thus decreases with the size of the main model. One challenge when scaling to larger and more recent models is the quality of the data the hypernetwork is trained on. A distillation objective on texts sampled from the LLM could help with this.
>
> > Do you think a similar approach would work for other modalities, such as multimodal models?
>
> Yes, applications to multimodal models could be a useful direction for future work. For example, the speech tokenization landscape might be even more heterogeneous than the text tokenization landscape, e.g. HuBERT (Hsu et al., 2021), AudioLM (Boros, 2022) and VALL-E (Wang et al., 2023) all use different ways of converting speech to discrete units. A ZeTT-style approach could help transfer across these different speech tokenizations. The same holds true for images, where ZeTT could be used e.g. to transfer to a different resolution of image patches, or across different image encoders. We will add these directions for future work to the paper.

---

> > ### Comment · Reviewer_1Y74 · 2024-08-10
> > **Thank you for your response and addressing my questions.**
> >
> > Looking for ward to see new results on latest LLMs, and multimodal expansion in the future.

---

### Official Review · Reviewer_EScf · 2024-07-06

**Soundness:** 3
**Presentation:** 3
**Contribution:** 3
**Rating:** 7
**Confidence:** 3

**Summary:**

The authors propose zero-shot tokenizer transfer, a new task where the goal is adapt language models with unseen tokenizers. They propose to tackle this problem by training a hypernetwork that can directly predict the embeddings for any given tokenizer, and such a hypernetwork is trained by sampling various tokenizations and minimizing a language modeling loss.

**Strengths:**

1. The authors propose an interesting task of zero-shot tokenizer transfer.
2. The authors have non-trivial designs for tokenizer sampling and the hypernetwork architecture.

**Weaknesses:**

1. Training the hypernetwork is time-consuming. However, as the authors mention, this is only a one-time cost.
2. The pretrained hypernetwork is specific to one particular LLM. As a result, this one-time cost needs to be paid for every LLM that wants to benefit from zero-shot tokenizer transfer. Practically, this may not be as efficient as simply retraining the embedding layer for each model.

**Questions:**

How long does it take for a trained hypernetwork to predict the embedding of a typical off-the-shelf tokenizer?

**Limitations:**

Appendix E and H.

---

> ### Author Rebuttal · Authors · 2024-08-06
>
> Thank you for your encouraging feedback!
>
> > The pretrained hypernetwork is specific to one particular LLM. As a result, this one-time cost needs to be paid for every LLM that wants to benefit from zero-shot tokenizer transfer. Practically, this may not be as efficient as simply retraining the embedding layer for each model.
>
> We made this design choice based on the observation that base LLMs have (relative, considering the pace of the field) longevity, while the number of possible tokenizers which transfer would be useful to is essentially unlimited. Due to this fact, training a hypernetwork quickly becomes more efficient than directly retraining the embedding layer, especially considering that our hypernetwork approach needs to see zero to ~1B tokens in the target tokenization for successful adaptation, while prior work typically needs hundreds of billions of tokens as shown by Dagan et al., 2024.
>
> > How long does it take for a trained hypernetwork to predict the embedding of a typical off-the-shelf tokenizer?
>
> Appendix J provides analysis of the required FLOPs (and thus the speed) of the hypernetwork. For Mistral-7B, the hypernetwork needs approximately 0.9% of the FLOPs of the base model per token. Thus, e.g. transferring to a tokenizer with 32k vocabulary size would take approximately as long as inference of 32k * 0.9% = 288 Mistral-7B tokens (scoring speed, not generation speed), which would take less than a second on most consumer-grade GPUs.

---

> > ### Comment · Reviewer_EScf · 2024-08-14
> >
> > Thanks for the response. I will maintain my score.

---

### Official Review · Reviewer_mdD9 · 2024-07-11

**Soundness:** 3
**Presentation:** 4
**Contribution:** 3
**Rating:** 6
**Confidence:** 4

**Summary:**

This paper presents a way to perform "zero-shot" construction of embeddings for a new target tokenizer. It proposes a hypernetwork based approach that learns to take in the embeddings of tokens generated by tokenizing the target token with the original tokenizer and produce the embedding of this token. The authors perform analysis on code and multilingual domains, and their method shows promise with respect to FOCUS, a recent embedding transfer method. Overall, the idea is novel, innovative, and seems promising. However, based on my understanding, I have significant concerns with the evaluation settings and some comparisons. I am willing to reconsider my assessment if my concerns on evaluations are adequately addressed.

**Strengths:**

* The approach is interesting, and the idea of using a hypernetwork to compute new embeddings is innovative.
* I can see this approach being useful as the tokenizers/vocabulary grows, since the number of parameters will stay constant. However I have some concerns on the ability to generalize (see weaknesses).

**Weaknesses:**

* I do not think ZETT is truly ‘zero-shot’. While it is important to make a distinction between vocabulary V and tokenization function T, it is likely that a large portion of embeddings are agnostic of T and more reliant on V. For example, I would not expect the embedding of the token ‘car’  to be much different when an LM is trained on two different tokenization functions.
* Therefore I think it is important to analyze the overlap of the distribution of the tokens. Appendix F analyzes the overlap with respect to the original vocabulary. But it is also important to study the overlap (both vocabulary and segmentations) between what UnigramLM generates during training versus the ‘target’ tokenizer during evaluation to really contextualize whether this approach is useful.
* The above wouldn’t be as big of a problem if the hypernetwork data did not contain the same languages/domains as that it was trained on. But from what I see all the evaluated languages are seen during hypernetwork training. I assume it will be something similar for code too.
* I think what would make more sense is to really evaluate on an ‘unseen’ language that uses the same script. Here one can expect a more pronounced distribution shift of both vocabulary and tokenization functions.

**Questions:**

* I cannot seem to find the value that was set for the hyperparameter max token length ‘l’. For a sufficiently large ‘l’ I would assume there would be a significant overlap of the vocabulary produced by UnigramLM during training and the target tokenizer during evaluation, especially when the domain/language is the same.
* When the target vocabulary size is increased, does the hypernetwork remain the same? If yes, I could see how some tokens might be zero shot transferred as the vocab size grows. However, without comparing the distributions as mentioned earlier, it’s hard to say what exactly the overlap is.
* How does ZETT compare to FOCUS when extra training is done in the more challenging multilingual setup? Would ZETT be a more attractive choice in this case compared to FOCUS which does not require any prior hypernetwork training?

**Limitations:**

I believe the weaknesses outlined above need to be addressed, which are not discussed. There is no separate limitations section in the paper as recommended in the checklist guidelines.

---

> ### Author Rebuttal · Authors · 2024-08-06
>
> Thank you for your feedback, and for recognizing the strengths of our approach.
>
> __Distinction between ZeTT and generalization to unseen tokens.__ The task we address is Zero-Shot Tokenizer Transfer since the hypernetwork and the base model have not seen the target tokenizer, that is, the specific combination of (V, T), which we want to transfer to. It is expected (and even desired) that there is vocabulary overlap between the tokenizers the hypernet (HN) has seen during training and the target tokenizer; this means that the HN should be able to generate good embeddings for these tokens. Our principal objective is (zero-shot and n-shot) generalization to unseen vocabularies and tokenization functions, not generalization to unseen tokens. Table 11 in Appendix I shows evidence toward successful generalization of this kind: the HN-predicted embeddings are more robust to different choices of tokenization function than off-the-shelf pretrained embeddings. Thank you for pointing out the ambiguity between ZeTT and generalization to unseen tokens in the current version of the paper. We will make sure to clarify the distinction.
>
> __Analyzing token distribution overlap.__ The above being said, we also fully recognize the importance of measuring and reporting generalization to unseen tokens and thank you for bringing this to our attention.
> To address this concern, we ran experiments to analyze the overlap of the distribution of tokens between the HN training tokenizers and the target tokenizers. We counted how often every token occurs in a training tokenizer during the entire HN training procedure, and used this to analyze how often the tokens in the different target tokenizers have been seen during training. The results are shown in Figure F1 in the attached PDF. We differentiate between tokens which occur in the evaluation data, and tokens which do not; this is important since the embeddings of tokens which do not occur in the evaluation data will not substantially impact performance. Notably, for XLM-R, >35% of occurring tokens in Greek, Bulgarian and Russian are unseen by the HN, even though the HN is trained on these languages. This is likely due to the non-Latin scripts. The hypernetwork still performs well in these languages with an average 2% performance decrease at 17% sequence length reduction on XNLI. In total, the HN has seen ~200M different tokens during training.
>
> __Extra Experiments with new Target Tokenizers.__ To test generalization to unseen tokens and out-of-distribution tokenizers, we have also conducted new experiments on (i) languages which are unseen by the HN but seen by the base model, (ii) languages which are unseen by both HN and base model and (iii) an out-of-distribution English word-level tokenizer. These results are available in the attached PDF.
>
> __Experiments on Unseen Languages.__ We ran additional experiments on XLM-R for two languages which are unseen by the HN but seen by the base model (Farsi and Dutch) and two languages which are unseen by both HN and base model (Aymara and Guarani). Results are shown in Table T1 in the attached PDF. The HN predicted embeddings still perform well in this case. Unseen languages do not necessarily have a high number of unseen tokens (as shown in Figure F1), likely due to the script being a confounding factor. However, for Aymara, where >40% of occurring tokens are unseen, the HN even outperforms the base model, at 36% reduced sequence length. This confirms that the HN generalizes to unseen tokens and languages.
>
> __Experiments on an English word-level tokenizer.__ To further evaluate generalization to out-of-distribution tokenizers, we transferred Mistral-7B to a tokenizer containing all words in the evaluation datasets (~100k words). The resulting model is thus word-level on our evaluation data. Results are shown in Table R2. As expected, there is a slight decrease in accuracy, but the gain over FOCUS persists, and sequence lengths are reduced by up to ~20%. 3.3k words are completely unseen in this setup and 13.5k words have been seen in less than 0.1% of training steps.
>
> __Additional Questions.__
>
> > the value that was set for the hyperparameter max token length ‘l’
>
> The max. sequence length `l` of the hypernetwork is shown in Table 7, it is 7 for English+Code hypernetworks and 15 for multilingual hypernetworks.
>
> > When the target vocabulary size is increased, does the hypernetwork remain the same?
>
> Yes, and we ran additional experiments to compare the distributions between training tokenizers and the target tokenizer. The new experiments show that transfer to large vocabulary sizes (for example, to word-level tokenization) is successful, and that the hypernetwork can generalize to unseen tokens.
>
>  > How does ZETT compare to FOCUS when extra training is done in the more challenging multilingual setup? Would ZETT be a more attractive choice in this case compared to FOCUS which does not require any prior hypernetwork training?
>
> Due to the duration of the author response period we were not able to run n-shot transfer experiments in a multilingual setup, however, due to the extent of the gap between FOCUS and the HN in the English+Code setup, we believe it is highly likely this gains of our method would persist in a multilingual n-shot setup.
>
> > There is no separate limitations section in the paper as recommended in the checklist guidelines.
>
> Thank you for pointing this out. We discuss limitations e.g. in Appendix E, F, and H. In case of acceptance we will use the extra page to add a dedicated limitations section with pointers to these appendices.

---

> > ### Comment · Reviewer_mdD9 · 2024-08-07
> >
> > Thank you for the detailed response. Most of my concerns have been addressed, and I am happy with the proposed revisions. I have increased my score.

---

### Author Rebuttal · Authors · 2024-08-06

We thank all reviewers for their feedback and reviews.

In response to the concerns about generalization to unseen tokens by reviewer mdD9 we have conducted additional experiments to quantify the overlap between the tokenizers seen during hypernetwork training and the target tokenizers. We also ran new experiments on unseen languages and on an out-of-distribution English word-level tokenizer, all of which exhibit positive results. The results are shown in the attached PDF. Please see the response to reviewer mdD9 for more details.

---

### Decision · Program_Chairs · 2024-09-25

**Decision:**

Accept (poster)

**Comment:**

Tokenizer transfer in general is a very practical and important problem. The paper clearly defines and motivates the ZeTT problem, highlighting its practical significance for cross-lingual and cross-domain applications, as well as for merging and ensembling models trained with different tokenizers. Further, this problem has not been explicitly addressed before to my knowledge, making this a valuable contribution to the NLP community. The paper achieved overall very positive reviews for evaluation, novelty and attention to details, with reviewers saying

Reviewer mdD9: "The authors perform analysis on code and multilingual domains, and their method shows promise with respect to FOCUS, a recent embedding transfer method."
Reviewer EScf: "The authors have non-trivial designs for tokenizer sampling and the hypernetwork architecture."
Reviewer 1Y74: "Especially impressive is the attention to the technical details described in the paper, for example, the text sampling strategy or the design of the appropriate loss function. Furthermore, the authors experiment with models of different architectures, different tokenizers, and on a variety of tasks, and verify the effectiveness of the proposed approach."

I therefore recommend accept.